# Knowledge, Attitudes, and Practice Regarding Physical Exercise in Type 2 Diabetic and Non-Diabetic Staff at a Tertiary Institution

**DOI:** 10.3390/ijerph21121707

**Published:** 2024-12-22

**Authors:** Mbuso Sibazo, Takshita Sookan-Kassie

**Affiliations:** Discipline of Biokinetics, Exercise and Leisure Sciences, School of Health Sciences, University of KwaZulu Natal, Durban 4041, South Africa; sookan@ukzn.ac.za

**Keywords:** type 2 diabetes, knowledge, attitudes, exercise adaptations

## Abstract

Type 2 diabetes mellitus (T2DM) has become a global epidemic, where increasing urbanization encourages sedentary lifestyles. Persistent physical inactivity can lead to T2DM and increase the risk of T2DM in the general population. Therefore, the aim of this study was to explore the knowledge, attitudes, and practices (KAP) regarding exercise amongst T2DM and non-diabetic (ND) staff at a tertiary institution in KwaZulu Natal South Africa. A total of 166 responses were received: a total of 16 responses (9.6%) were T2DM, and 150 responses (90.0%) were non-diabetic (ND). The demographics included 66.3% females and 33.7% males who consented to taking part, 62.7% were black, 18.7% were Indian, 12% were white, 5.4% were colored, and 1.2% were other. A cross-sectional descriptive survey design, utilizing a modified validated online knowledge, attitudes, and practice questionnaire, was used to collect data. Descriptive statistics were used for the analysis: inferential statistics; the ordinal (1–5) Likert scale; *t*-tests; and chi-square tests. The level of statistical significance was set at *p* ≤ 0.05. No significant differences were found between the T2DM and ND groups except in their attitude towards exercise, which showed three items with significant differences. The ND group agreed significantly more than the T2DM group that they looked forward to exercising (*p* = 0.002), and even without company, they exercised regularly (*p* = 0.042). The T2DM group agreed significantly more with the statement that they had asked their doctor if there was medicine available to make them better without doing any exercise (*p* = 0.002). The overall KAP results of the current study found that both participants diagnosed with T2DM and those in the ND group know about exercise and have a good attitude toward exercise. However, both groups still have poor practice regarding physical activity.

## 1. Introduction

Type 2 diabetes mellitus (T2DM) has become a global epidemic, especially in developing countries where increasing urbanization encourages a sedentary lifestyle [1,2] causing a rise in morbidity and mortality. The worldwide increase in sedentary lifestyles, aging, and obese populations has greatly contributed to the swift rise in T2DM occurrence and prevalence [3,4]. Epidemiological studies have revealed that obesity is the most crucial risk factor for T2DM, with almost 90% of the patients developing T2DM as a result of excessive body weight [4,5].

Type 2 diabetes mellitus is a chronic metabolic disorder [6] resulting from a combination of the inability of the muscle cells to react to insulin (insulin resistance) and inadequate compensatory insulin secretion [7,8]. This, then, results in increased concentrations of glucose in the blood, which is known as hyperglycemia. The development of T2DM is greatly influenced by a wide range of factors, including physical inactivity; lifestyle habits (consumption of a high-sugar diet and a sedentary lifestyle); increasing age; smoking; alcohol consumption; and genetics [5,9,10,11]. According to the World Health Organization, nearly 90% of all T2DM cases develop the disease as a result of excess body weight. Numerous studies have demonstrated that a steady increase in body weight over time significantly escalates the chances of T2DM development [12]. The research clearly shows that diabetes can be prevented through lifestyle changes [1,13].

The literature has also shown that the general population, due to increasing urbanization, is less active, despite the proven benefits of exercise. This is especially important for aging individuals predisposed to diseases resulting from increasing age, which comes with numerous complications [14]. Randomized controlled trials and prospective epidemiological data provide strong and consistent evidence that moderate-to-vigorous physical activity reduces the risk of cardiovascular diseases, T2DM, metabolic syndrome, and other conditions [15]. However, large percentages of the world’s population remain inactive. Recent data from 168 countries indicate that 27.5% of adults are not active enough to achieve health benefits, and these numbers are doubled in high-income countries compared to low-income countries [14]. A study conducted in South Africa found that a high percentage of older adults (50 years and above) did not engage in vigorous-or-moderate physical activity [16,17].

Over the past years, researchers have investigated the knowledge, attitudes, and practice (KAP) regarding exercise of individuals with T2DM, and varying results have been reported [18,19,20,21,22]. Despite studies that have investigated the benefits of exercise for both the general population and individuals with T2DM, there is still limited literature on whether the KAP regarding exercise are connected with the under-participation in exercise in both these populations. Therefore, the research question is as follows: What is the knowledge, attitudes, and practice regarding exercise of individuals diagnosed with T2DM? The aim of this study was to explore the KAP of T2DM and ND staff of the University of KwaZulu Natal (UKZN) in South Africa.

## 2. Materials and Methods

### 2.1. Study Population, Sample Size, and Sampling

This study adopted a cross-sectional descriptive survey design to explore a specified population’s current opinions and practices. This study’s sample was selected by utilizing a convenience sampling technique, and the staff from the UKZN from all five campuses were recruited to participate. The participants that were included were those diagnosed with T2DM (those on oral medication and insulin-dependent), ND staff, and those who consented to taking part. The exclusion criteria consisted of participants who did not consent, non-UKZN staff, and individuals diagnosed with type 1 or gestational diabetes.

### 2.2. Procedure

A structured questionnaire created from existing validated questionnaires was used to collect data. The questionnaire included the Knowledge Questionnaire [23] and the Exercise Attitude Questionnaire-18 [24]. The Exercise Attitude Questionnaire-18 is regarded as an easy-to-use, portable instrument that investigates attitudes to exercise [24]. It is easy to understand; is not time consuming; has good validity; and has been appreciated by both professionals and patients (Manigandan et al., 2004), and the International Physical Activity Questionnaire [23]. The questionnaire consisted of two sections: Section One covered demographics and the methods used to manage type 2 diabetes; and Section Two looked at knowledge, attitudes, and practice regarding exercise. The knowledge sub-section had ten questions, with responses to the first four questions either true, false, or not sure. The responses to the last six questions were ranked on a Likert-scale type as ‘strongly disagree’, ‘disagree’, ‘neutral’, ‘agree’, and ‘strongly agree’. The responses to the 18 questions on attitudes were also ranked on a Likert-scale type. The practice sub-section required the participants to provide the number of times per week they performed the mentioned exercises, categorized as strenuous, moderate, or mild/light exercise.

### 2.3. Ethical Consideration

Ethical clearance was requested and granted from the University of KwaZulu-Natal’s Biomedical Research Ethics Committee, with the protocol reference number (BREC/00002992/2021). Gatekeeper’s permission was obtained from the University of KwaZulu-Natal’s registrar.

### 2.4. Data Analysis

The data from the questionnaire were captured in Microsoft^®^ Excel 2010, exported to SPSS version 25, and analyzed. Descriptive statistics, including means and standard deviations, were calculated where applicable to summarize the data. Frequencies and percentages were used for categorical data. The binomial test was used to determine whether a significant number of the participants had a family history of diabetes or not. The binomial test was also used on the knowledge questions to test if a significant proportion of the participants had any of the knowledge items correct. The comparison between the T2DM and ND groups was achieved using Pearson’s chi-square test. The independent samples *t*-test was used to determine if the average mark/percentage was significantly different between the T2DM and ND groups.

## 3. Results

### 3.1. Demographic Characteristics of the Participants

A total of 166 respondents participated in this study: a total of 66.3% respondents were females, and 33.7% were males; a total of 62.7% respondents were black, 18.7% were Indian, 12% were white, 5.4% were colored, and 1.2% were other. Most of the participants were aged between 30 and 39 years (31.3%), while participants younger than 30 years old accounted for 27.7% of the sample. Of the 166 participants, 16 had T2DM, and 150 were ND. Several ND participants did not give answers as asked in the final part of the questionnaire (practice regarding exercise), so they were not included in the analysis of that section. Table 1 summarizes the demographic data.

### 3.2. Knowledge Results with Regards to Exercise in T2DM and ND

The mean score for knowledge about exercise for the whole sample (N = 166) was 64.20% ± 25.80. The binomial test was used to test whether a significant proportion of the total sample (N = 166) answered any of the questions about exercise correctly. A significant proportion (59.6%) knew that physical activity and exercise are not the same thing (*p* = 0.016). In terms of knowing whether people with T2DM can safely perform an exercise, 72% of the participants knew this to be true (*p* < 0.001). Of the participants, 80% knew that exercise can be used in the management/treatment of T2DM (*p* < 0.001). It was also found that 90.9% understood that T2DM management should include both exercise and a healthy diet (*p* < 0.001). A statistically significant proportion (69.2%) of the participants knew that exercise does not have a negative effect on the control of T2DM (*p* < 0.001). Furthermore, 62% of the participants showed that they were aware that early detection of excessive weight and inactivity can delay or prevent T2DM (*p* = 0.002). Table 2 summarizes the frequency with which each item was chosen by the participants and includes the *p*-value.

### 3.3. Comparison Between the T2DM and ND Groups of Their Knowledge Regarding Exercise

No significant difference (*p* = 0.280) was discovered between the T2DM and ND groups in terms of their overall knowledge regarding exercise. Table 3 shows the results obtained when comparing the T2DM group and the ND group on their knowledge regarding exercise. Using Pearson’s chi-square test, a significant relationship was discovered between having T2DM and knowing whether T2DM patients can do strenuous exercises, like weightlifting and running (*p* = 0.031). A significant number in the T2DM group knew the answer to this. A significant relationship (*p* = *0*.027) was also found between having diabetes and knowing that detecting excessive weight and inactivity early can delay or prevent T2DM.

### 3.4. Attitudes to Exercise in the T2DM and ND Groups

Table 4 summarizes the results from the one-sample *t*-test. There is significant agreement that they would continue to exercise for the good of their health (*p* = 0.001); they believed that their health and fitness would improve with exercising (*p* = 0.001); they looked forward to exercising (*p* = 0.001); they felt that age was an influencing factor in motivating them to exercise (*p* = 0.001); even without company they exercised regularly (*p* = 0.001); and they were strict about doing regular exercise, as it kept them alert and energetic (*p* = 0.001).

There was significant disagreement that their regular work was an adequate substitute to exercise (*p* = 0.001); they used mild pain or fatigue as an excuse to keep away from exercising (*p* = 0.001); exercising took away most of their energy as they were already feeling weak and exhausted (*p* = 0.001); they felt embarrassed exercising in front of others (*p* = 0.001); they would rather suffer with their problems than exercise (*p* = 0.001); they exercised to satisfy their families (*p* = 0.001); and they thought of asking their doctors if there were any medicines available which would have made them better without doing exercise (*p* = 0.001).

### 3.5. Comparison Between the T2DM and ND Groups on Their Attitudes to Exercise

A comparison between the T2DM group and ND group to determine whether a significant difference existed between them was accomplished using an independent sample *t*-test. Most of the attitude items did not reveal significant differences between the two groups; only three items showed significant differences. The ND group agreed significantly more than the diabetics that they look forward to exercising. The results also showed that the ND group agreed significantly more that, even without company, they exercised regularly. It was also seen that the diabetics’ group agreed significantly more that they had asked their doctor if there were any medicines available that would make them better without doing any exercise.

### 3.6. Exercise Practices of the T2DM and ND Groups

Of the 150 ND participants, several of the participants (N = 17) were not included in the analysis. The mean score for the whole sample (N = 149) was 33.30 ± 44.50. As shown in the table below (Table 5), the mean score for the T2DM sample was 26.88 ± 26.11, and for the ND sample, it was 34.07 ± 46.23. The T2DM and ND groups did not show any significant difference in terms of their weekly activity score.

## 4. Discussion

Type 2 diabetes mellitus statistics in South Africa in 2021 were reported in one study to be 15.25%, and for KwaZulu-Natal, the statistics were 17.36% [25]. Another study published in 2021, looking specifically at Durban, found that, in individuals aged 45 years and younger, 10.70% of them are type 2 diabetic, while 12.84% of individuals aged 55–64 years are type 2 diabetic [26]. The age range of the current study is between 30 and 69 years, and there is at least one participant with T2DM in the sample who is in each age category [26]. The current study received a response of 9.6% (nearly 10%) of individuals with T2DM, which is similar to the above-mentioned studies. Therefore, the sample obtained is approximately representative of the population of Durban where the present study was conducted.

This study looked at the knowledge, attitudes, and practice (KAP) regarding exercise and found that both T2DM and ND groups had good knowledge about exercise, answering 66.67% of the knowledge items correctly. Furthermore, there were no significant differences between the T2DM group and the ND group in their overall knowledge regarding exercise. These findings could be attributed to the fact that the study sample was drawn from university staff as well as the urban location of the participants. This study also showed that more than half of the participants had a family history of T2DM (62.5%), explained by the familial prevalence of T2DM. In addition, there is a high prevalence of T2DM in the eThekwini district, which could have led to the high public awareness of the disease [27]. Similarly, Sookan et al. (2022) reported that 94.7% of T2DM patients in Durban had good knowledge regarding physical exercise. In another study, looking at insights and behavior of the general population regarding modifiable risk factors for the prevention of type 2 diabetes mellitus, it was reported that the general public had sufficient knowledge about exercise as a preventative measure for T2DM [28]. One study, conducted in Kilimanjaro, Northern Tanzania, assessing the knowledge, attitudes, and practices regarding physical exercise, reported that 98.4% of the participants had high levels of knowledge regarding physical exercise [29].

However, contradicting results regarding the level of knowledge of the type 2 diabetic population and the general population have been noted in the literature. A study conducted in King Abdullah Hospital, Bisha, in 2021 found that 57.5% of the participants did not have sufficient knowledge of the role of exercise in the management of type 2 diabetes [17]. These results were seen in two other studies conducted in Libya and Zimbabwe, where the populations showed poor knowledge regarding exercise [20,30,31].

The present study showed that there were no significant differences (*p* = 0.280) between the T2DM group and the general population group regarding their knowledge about exercise. It is crucial for the general population to be knowledgeable about exercise, as the major risk factors that lead to T2DM are physical inactivity and being overweight and obesity [32]. The participants of the present study showed that they were aware of the impact of excessive weight and physical inactivity (*p* = 0.002). The significance of being adequately informed about major T2DM risk factors to prevent the disease and the urban location and working environment of the participants could possibly have resulted in no differences between the T2DM and ND groups in their knowledge regarding exercise.

One of the factors that can fuel motivation for a certain behavior can be one’s attitudes to that behavior [33]. The present study investigated the attitudes of both T2DM patients and the ND group to exercise as a means of managing and/or preventing T2DM, and it was found that both groups had positive attitudes to exercise. These results are consistent with what has been seen in numerous studies looking at attitudes to exercise in individuals with type 2 diabetes and the general population [19,29,34,35]. Another study conducted in Bloemfontein, South Africa, to determine the attitudes to lifestyle modification for individuals diagnosed with T2DM reported that the individuals with T2DM had a positive attitude to exercise [36]. Similar results were found in other studies [37,38,39,40].

In a study conducted on older African American adult, Asian, and Hispanic populations to investigate the perceptions, opinions, beliefs, and attitudes regarding exercise and physical activity, it was observed that the older adults had positive attitudes to exercise [41,42]. Similar results showing positive attitudes to exercise have been noted in other studies [20,43,44,45].

The T2DM and ND groups were compared to ascertain whether there were any significant differences in their attitudes to exercise. Most of the attitude items did not reveal any significant differences between the two groups; only three items showed significant differences. There were positive results for both groups, and the small differences could be attributed to the considerable knowledge that both groups have about exercise, as they are aware of the benefits of exercise, both for those diagnosed with T2DM and for the general population [46,47]. The three items that the two groups differed on were as follows: The ND group agreed significantly more than the T2DM group that they look forward to exercising; the NDs agreed significantly more that, even without company, they exercised regularly; and, lastly, the T2DM group agreed significantly more that they had asked their doctor if there were any medicines available that would make them better without doing any exercise.

It is believed that good knowledge of, and positive attitudes to, exercise will produce favorable exercise behavior in both individuals suffering from T2DM and the general population [48]. However, the current study produced rather less promising results, as it was found that the overall study sample did not meet the recommended weekly requirement for physical exercise (22.30 ± 44.50), which is 150 min of moderate-intensity exercise [46]. The researchers noted these results, since similar evidence of poor practice by those diagnosed with T2DM and the general population has been noted in numerous studies [6,37,40]. Aarsha et al. (2022) conducted a research study to investigate the KAP regarding physical exercise, and it was found that the majority of the participants (86.8%) had good knowledge of, and a positive attitude to, exercise, yet presented with poor practices regarding exercise. A similar research study conducted in Singapore on the impact of knowledge and attitude to lifestyle practices in the prevention of T2DM observed that the participants had good knowledge regarding physical exercise and also presented positive attitudes to exercise; however, 72.2% of the sample did not meet the Health Promotion Board’s physical activity recommendation [6,40].

Exercise has numerous benefits for individuals diagnosed with T2DM [12,14,49], not just for the maintenance of weight and blood glucose but also providing physical, social, emotional, and cognitive benefits [48]. When considering that these individuals, both T2DM patients and the general population, have good knowledge of, and positive attitudes to, exercise, as seen in the present study and other studies [18,34,35,40], researchers need to consider the reasons and barriers that lead to poor practice.

Individuals suffering from T2DM are at risk of developing other comorbidities [50,51,52]. Comparing the two groups, the T2DM group and the general population in the present study regarding their engagement in physical exercise, there were no significant differences between the groups. Other studies [53,54,55,56,57] have revealed a similar trend in reported barriers to exercise.

## 5. Conclusions

The overall KAP results of the current study found that patients diagnosed with T2DM and the ND group are well informed about exercise and have a good attitude to exercise. However, they still display poor habits regarding exercise practice. Educational programs need to be implemented to educate and encourage individuals, both those diagnosed with T2DM and those without T2DM, to engage in exercise.

## Figures and Tables

**Table 1 ijerph-21-01707-t001:** Characteristics of participants (N = 166).

Variables	Frequency (n)	Percentage
Gender:		
Male	56	33.7%
Female	110	66.3%
Age years:		
<30	46	27.7%
30–39	52	31.3%
40–49	34	20.5%
50–59	26	15.7%
60–69	8	4.8%
Race:		
Black	104	62.7%
White	20	12.0%
Indian	31	18.7%
Colored	9	5.4%
Other	2	1.2%
Suffer from T2DM:		
Yes	16	9.6%
No	150	90.4%

**Table 2 ijerph-21-01707-t002:** Summary of the frequency of responses to each item of the combined sample.

Items	Frequency (%)		n	*p*-Value
	Incorrect	Correct		
**1. Physical activity and exercise are the same thing.**	**67 (40.4)**	** *99 (59.6)* **	166	** *0.016 ** **
2. People with T2DM can safely perform exercise.	47 (28.3)	** *119 (71.6)* **	166	***<0.001*** *
3. Exercise can be used in the management/treatment of diabetes.	33 (19.8)	** *133 (80.1)* **	166	***<0.001*** *
4. T2DM cannot do strenuous exercise like weightlifting, cycling, running.	84 (50.6)	82 (49.3)	166	0.938
5. Exercise is as effective as medication in managing T2DM.	87 (52.4)	79 (47.5)	166	0.587
6. Diabetes management should include both exercise and a healthy diet.	15 (9.0)	** *151 (90.9)* **	166	***<0.001*** *
7. Exercise prescribed by anyone can be used to manage T2DM.	88 (53.0)	78 (46.9)	166	0.485
8. Exercise has a negative effect on the control of T2DM.	51 (30.7)	** *115 (69.2)* **	166	***<0.001*** *
9. Early detection of excessive weight and inactivity can delay or prevent T2DM.	63 (37.9)	** *103 (62.0)* **	166	***0.002*** *

* Based on Z approximation. A *p*-value given as 0.000 is very small and reported as *p* < *0*.001; a *p*-value of, e.g., 0.017, is reported as *p* = *0*.017. The binominal test was used to see whether a significant number of participants (N = 166) had any items correct.

**Table 3 ijerph-21-01707-t003:** The table shows the comparison of the knowledge results using Pearson’s chi-square test between the T2DM and the ND groups.

Items	Diabetic	Responses as Frequency (%)		Χ^2 /^ Fisher’s Exact	Df	*p*-Value
		Incorrect	Correct			
**Physical activity and exercise are the same thing.**	**No**	**61 (40.7)**	**89 (59.3)**	0.60	1	0.806
	Yes	6 (37.5)	10 (62.5)			
People with T2DM can safely perform exercise.	No	43 (28.7)	107 (71.3)	0.96	1	0.757
	Yes	4 (25)	12 (75)			
Exercise can be used in the management/treatment of diabetes.	No	30 (20)	120 (80)	0.14	1	0.905
	Yes	3 (18.8)	13 (81.3)			
T2DM cannot do strenuous exercise like weightlifting, cycling, running.	No	80 (53.3)	70 (46.7)	** *46.43* **	** *1* **	***0.031*** *
	Yes	4 (25)	** *12 (75)* **			
Exercise is as effective as medication in managing T2DM.	No	77 (51.3)	73 (48.7)	7.23	1	0.395
	Yes	10 (62.5)	6 (37.5)			
Diabetes management should include both exercise and a healthy diet.	No	13 (8.7)	137 (91.3)	2.58	1	0.611
	Yes	2 (12.5)	14 (87.5)			
Exercise prescribed by anyone can be used to manage T2DM.	No	81 (54)	69 (46)	6.10	1	0.435
	Yes	7 (43.8)	9 (56.3)			
Exercise has negative effects on the control of T2DM.	No	47 (31.3)	103 (68.7)	2.72	1	0.778
	Yes	4 (25)	12 (75)			
Early detection of excessive weight and inactivity can delay or prevent T2DM.	No	61 (40.7)	89 (59.3)	** *48.71* **	** *1* **	***0.027*** *
	Yes	2 (12.5)	** *14 (87.5)* **			

* In place of the chi-square test, to see whether a significant relationship exists between the T2DM and ND groups regarding their knowledge, the Fisher’s exact test was used, as the conditions were not met for the chi-square test due to the small sample size of the T2DM group.

**Table 4 ijerph-21-01707-t004:** Table showing results of the T2DM and ND groups in their attitudes towards exercise using a one-sample *t*-test (N = 166).

Items	Responses as Frequency (%)					n	Mean (±SD)	*p*-Value
	Strongly Disagree	Disagree	Neutral	Agree	Strongly Agree			
**I feel that my regular work is an adequate substitute for exercise.**	**61 (36.7)**	**63 (38.0)**	14 (8.4)	17 (10.2)	11 (6.6)	166	2.12 (1.205)	<0.001 *
I need someone to keep prompting me to exercise.	21 (12.7)	48 (28.9)	22 (13.3)	52 (31.3)	23 (13.9)	166	3.05 (1.292)	0.632
I use mild pain or fatigue as an excuse to keep away from exercising.	31 (18.7)	63 (38)	32 (19.3)	35 (21.1)	5 (3)	166	2.52 (1.110)	<0.001 *
I feel exercising takes away most of my energy, as I am already feeling weak and exhausted.	45 (27.1)	84 (50.6)	16 (9.6)	18 (10.8)	3 (1.8)	166	2.10 (0.980)	<0.001 *
I will continue to exercise for the good of my health.	6 (3.6)	7 (4.2)	24 (14.5)	68 (41)	61 (36.7)	166	4.03 (1.006)	<0.001 *
I believe I will definitely improve my health and fitness with exercising, as I have seen others improving.	8 (4.8)	3 (1.8)	19 (11.4)	80 (48.2)	56 (33.7)	166	4.04 (0.981)	<0.001 *
I look forward to exercising.	8 (4.8)	22 (13.3)	30 (18.1)	80 (48.2)	26 (15.7)	166	3.57 (1.058)	<0.001 *
I feel that age is an influencing factor in motivating me to exercise.	13 (7.8)	32 (19.3)	33 (19.9)	66 (39.8)	22 913.3)	166	3.31 (1.159)	0.001 *
I feel embarrassed exercising in front of others.	36 (21.7)	64 (38.6)	33 (19.9)	17 (10.2)	16 (9.6)	166	2.48 (1.215)	<0.001 *
Even without company I exercise regularly.	14 (8.4)	33 (19.9)	32 (19.3)	56 (33.7)	31 (18.7)	166	3.34 (1.230)	<0.001 *
I feel that I have no time of my own and exercising takes away my valuable time.	29 (17.5)	66 (39.8)	36 (21.7)	28 (16.9)	7 (4.2)	166	2.51 (1.094)	<0.001 *
I would rather suffer with my problems than exercise.	74 (44.6)	69 (41.6)	18 (10.8)	3 (1.8)	2 (1.2)	166	1.73 (0.818)	<0.001 *
I exercise to satisfy my family.	67 (40.4)	75 (45.2)	16 (9.6)	5 (3.0)	3 (1.8)	166	1.81 (0.866)	<0.001 *
I feel that people are making false claims when they explain about the benefits of exercising.	48 (28.9)	57 (34.3)	43 (25.9)	14 (8.4)	4 (2.4)	166	2.21 (1.032)	<0.001 *
I thought of asking my doctor if there are any medicines available which will make me better without doing exercise.	55 (33.1)	57 (34.3)	31 (18.7)	19 (11.4)	4 (2.4)	166	2.16 (1.084)	<0.001 *
I am strict about doing regular exercise as it keeps me alert and energetic.	14 (8.4)	33 (19.9)	41 (24.7)	54 (32.5)	24 (14.5)	166	3.25 (1.178)	<0.008 *

* indicates significance at the 95% level (*p* = 0.05). The one-sample *t*-test was used to test for significant differences in the average scores from the central score (N = 166).

**Table 5 ijerph-21-01707-t005:** Summary of the descriptive statistics and the results from the independent sample test.

	Category	n	Mean (±SD)	t	Df	*p*-Value
Activity score	Diabetic	16	26.88 (26.110)	0.610	147	0.543
	ND	133	34.07 (46.231)			

## Data Availability

The raw data supporting the conclusions of this article will be made available by the authors on request. A transcribed version of the data can be accessed from https://docs.google.com/forms/d/1-Ry3taSYTNCXeAVi36ZMjb0CBPB1Xlu-TMQBvfKHoA8/edit?usp=forms_home&ths=true (accessed on 9 April 2021).

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
