# Peer review of "Knowledge, Attitudes, and Practice Regarding Physical Exercise in Type 2 Diabetic and Non-Diabetic Staff at a Tertiary Institution"

_ijerph, 2024, doi:10.3390/ijerph21121707_

Round 1
Reviewer 1 Report
Comments and Suggestions for Authors
General Comments: Your manuscript covers an interesting topic but does have some issues you need to address.
Specific Recommendations:
Page 2
Line 54 – You need to add “of” between “percentage” and “older” and change “adult to adults” as shown below.
“A study conducted in South Africa found that a high percentage of older adults (50 years and above) did not engage in vigorous-or-moderate physical activity [16,17].
Line 65 – 2.1 under Materials and Methods
Study Population, Sample Size and Sampling.
Combine the first sentence to the rest of this section and eliminate the Study population on line 68 as shown below.
“The study adopted a cross-sectional descriptive survey design to explore a specified population’s current opinions and practices. The study sample was selected by utilizing a convenience sampling technique and the staff from the UKZN from all five campuses were recruited to participate. Participants included T2DM staff (those on oral medication and insulin-dependent), ND staff and those who consented to taking part. Participants who did not consent, non-UKZN staff and individuals diagnosed with type 1 or gestational diabetes were excluded from the study.”
Page 4 Results – You give the readers a lot of information concerning the overall difference between T2DM and ND subjects but do not give the composition of each group. In other woods what was the age, sex, and race of each group? With the T2DM group being only 16 in comparison to the 1010 of the ND group it could have a major effect. It is something you need to include.
Page 6 Figure 1 – Eliminate figure 1. It provides the same data as the tables and is not needed.
Reviewer 2 Report
Comments and Suggestions for Authors
Knowledge, Attitude, and Practice Regarding Exercise in Type 2 Diabetic and Non-Diabetic Staff at a tertiary institution REVIEW
Regarding title, although I’m not native English speaker and authority in the field of language I recommend usage of phrase “Attitudes” instead “Attitude” and also clarification of term “Exercise” such as “Physical Exercise” or something similar.
This study aimed to explore the knowledge, attitudes, and practices regarding exercise amongst type 2 diabetes mellitus and non-diabetic at a tertiary institution. Having relatively large and specific total sample is studies major strength. On the other hand, small T2DM group for cross-sectional design is studies main weakens, which which compromises the fulfillment of the main goal of the study.
The abstract lacks with information about the sample and questioner. Also, lines 13, 14, 15, needs rework since it is not clear stated.
The introduction provides a clear basis for setting the research question, which is stated explicitly. Jet, introduction lacks hypotheses.
Line 60/61 “there is still limited literature on whether KAP of exercise is connected with the under-participation in exercise” is a trivial observation. This cannot be a justification for the investigations carried out, and it should be elaborated.
Lines 80/83 needs rework.
Sampling technique, inclusion and exclusion criteria are not explicated enough. Also it is not clear how authors determine sample size. It would be also beneficial for readers if authors have to share some quantitative data about the sample (body mass, body high, BMI and similar). Method section lacks of power analyses and effect size explanation in data processing part.
It is not clear whether the preconditions of normality of distribution and homogeneity of variance are met, which should be explained in the method and results. This will give us whether appropriate data analysis was carried out. Analysis of variance was used instead of the t test. If the previous statement is ignored, it can be said that results are properly interpreted, and appropriate conclusions were drawn.
The references are appropriate.
I have not additional comments.
Round 2
Reviewer 1 Report
Comments and Suggestions for Authors
You have made appropriate changes.
Reviewer 2 Report
Comments and Suggestions for Authors
Tahnks for cooperation.